# RWR-RGCN : A Novel Framework for Fraud Detection via Node Context Aggregation

## Abstract

The integrity of online reviews is crucial for businesses, yet widespread review fraud poses significant risks. This paper addresses this challenge by leveraging the power of multi-relational graph convolutional networks (RGCNs) for fraud detection. We introduce RWR-RGCN, a novel framework integrating a multi-layer RGCN architecture with Random Walks with Restart (RWR). The essential role of capturing critical connections lies in RWR generating node sequences, which can aggregate node features, enhancing the model's understanding of the local and global context within the review graph. To further refine fraud detection, we incorporate Louvain clustering for community identification, identifying high-modularity clusters indicative of coordinated fraudulent activity. Evaluated on the Yelp dataset, RWR-RGCN achieved a recall of 94.56%, surpassing the state-of-the-art and baseline methods. These results demonstrate the superior effectiveness of the proposed framework in detecting fraudulent activity within complex online review networks.

## 1 Introduction

Online review fraud increasingly threatens platform integrity, as fraudsters adeptly mask their actions by establishing authentic links with regular users, a tactic that leads conventional graph-based detectors to acquire non-discriminative embeddings. Identifying disguised fraud in multi-relational graphs necessitates the concurrent acquisition of local behavioral signals, global community structure, and relation-specific interaction patternscapabilities that no current approach offers collectively.

Graph Neural Networks have made significant contributions to fraud detection in financial crimes (Kurshan et al., 2020), (Kurshan & Shen, 2020), fake news (Li & Li, 2024), (Xu et al., 2022), opinion fraud (Liu et al., 2020), (Li et al., 2019), and healthcare fraud Zhang et al. (2024). In contrast to traditional graph-based methods, GNN-based approaches derive the representations of nodes by end-to-end aggregation of neighboring features, therefore minimizing costs associated with feature engineering and data annotation (Hamilton et al., 2017) while effectively capturing complex relationship patterns (Zhang et al., 2024).

Nevertheless, despite these advancements, GNN-based fraud detectors are impeded by two fundamental challenges: class imbalance, characterized by a minority of fraudulent nodes, and heterophily, wherein fraudulent nodes often establish connections with legal ones Schlichtkrull et al. (2018). Specialized models tackle these issues from various perspectives: GraphConsis Liu et al. (2020) enforces neighbor consistency; CARE-GNN Dou et al. (2020) employs adaptive neighbor selection; PC-GNN Liu et al. (2021) resamples minority nodes; FRAUDRE Zhang et al. (2021) utilizes relational modeling for inconsistent graphs; RioGNN Peng et al. (2021) unravels homophilic and heterophilic relational signals; and DOS-GNN Jing et al. (2024) integrates dual-feature aggregation with embedding-space oversampling to address both challenges concurrently.

In Dou et al. (2020) demonstrates the two categories of camouflage: as Li et al. (2019) first features types, cunning con artists can modify their actions, utilize language generation models, or insert special characters into reviews (a tactic known as spamouflage) to mask overtly questionable consequences to facilitate eluding feature-based detectors. The second relation type is when the graph has more benign nodes than fraudsters. Fraudsters can investigate the defenders' graphs and modify their actions to reduce suspicion (Zhuang et al.,

2024). More popular camouflage techniques include putting the poster in contact with high-credit users, eliminating critical remarks, and fabricating supportive comments (Yang et al., 2021).

GNNs aggregate features from surrounding nodes to power graphs. This method capitalizes on data homophily, as similar nodes are more likely to be connected (Gong et al., 2023). While this approach enhances learning through accurate predictions, it also has a drawback: oversmoothing (Liu et al., 2020). As hierarchically aggregated information from surrounding nodes, graph nodes may lose their particular qualities and become indistinguishable. Fraudsters typically hide in networks, making this issue troublesome for fraud detection. According to (Liu et al., 2020), "camouflage fraudsters" use graph homophily, making it difficult for GNNs to distinguish between genuine and fraudulent nodes. The oversmoothing effect masks even more minor differences needed to identify fraud. Thus, while neighbor aggregation in GNNs is useful, it must be regulated to avoid enabling fraudsters hide.

Current methodologies tackle these issues independently, resulting in a significant deficiency. CARE-GNN Dou et al. (2020) employs reinforcement learning for adaptive neighbor selection, functioning solely at the node-to-node level; it can identify fraudulent individual connections but is incapable of detecting collusive fraud networks that appear solely at the community level. PC-GNN Liu et al. (2021) mitigates class imbalance by resampling but lacks an approach for disguise identification or group-level structural analysis. RioGNN Peng et al. (2021) attains a robust AUC via relation-aware disentanglement; nonetheless, its recall of 75.08% indicates that it still overlooks a significant number of fraudstersparticularly those whose individual node attributes seem typical but whose community affiliation shows collaboration. DOS-GNN Jing et al. (2024) addresses heterophily and imbalance by aggregation and oversampling; yet, it fails to simulate community-level fraud structures entirely.

This research introduces RWR-RGCN, a framework based on the claim that disguises, collusion, heterophily, and class imbalance are interrelated issues necessitating three complementing signals functioning sequentially. Initially, Random Walk with Restart produces distinctive features from the surrounding nodes that reveal behavioral discrepancies between fraudsters and legitimate users at the path level, effectively addressing feature-level obfuscation. Secondly, Louvain community identification divides the graph into high-modularity communities, maintaining cohesive fraud connections as singular structural entities that remain identifiable even when individual intra-ring connections seem normal  a feature lacking in all previous node-level approaches. Third, RGCN executes feature propagation, effectively capturing the unique fraud signals associated with each relation. In contrast to CARE-GNN Dou et al. (2020) and RioGNN Peng et al. (2021), the framework does not necessitate reinforcement learning. In contrast to PC-GNN Liu et al. (2021) and DOS-GNN Jing et al. (2024), it necessitates no synthetic oversampling. The outcome is a streamlined, interpretable pipeline that shows enhancement over RioGNN Peng et al. (2021) by concurrently tackling aspects that no previous technique has addressed collectively. The proposed system provides a scalable, interpretable, and effective solution to combat the evolving evasion strategies employed by fraudsters.

The remainder of this paper is structured as follows: Section 2 provides problem formulation, Section 3 provides background and related work. The methods we propose are outlined in Section 4. Section 5 presents the performance evaluation and results. Finally, we provide a discussion in Section 6 and a conclusion and future work in Section 7.

## 2 Preliminaries and Problem Formulation

### 2.1 Graph Definition

An undirected graph $\mathcal{G} = (V, E)$ is defined as a set of nodes or vertices $V$ and a set of links or edges $E \subseteq V \times V$ that link the nodes together. Any two nodes $i$ and $j$ are adjacent if they are connected with a link. Edges are defined as $(u, v)$, and it is supposed that the order of nodes in the pair does not matter. $\boldsymbol{A}$ an adjacency matrix, which is an $n \times n$ with $n = |V|$, is a popular method of representing a graph. If there is an edge between node $i$ and node $j$, it is indicated by the $(i, j)$ entry of the matrix $\mathbf{A}_{i,j}$. An edge's weight can be represented by the corresponding item in the adjacency matrix and can be either a real integer for a weighted graph or a binary value (0 or 1) for an unweighted graph. Both directed and undirected edges are possible. Directed edges contain information that has a direction; for instance, a road could be a one-way

street. The geographical distance between two weather stations is an example of an undirected edge, which has no source.

Formally, an ordered tuple $(u, v)$ represents a directed edge $e \in E$ between nodes $u \in V$ and $v \in V$, while the unordered variant $\{u, v\}$ represents an undirected edge. Nodes and edges can also have a feature vector $\boldsymbol{a} = (a_1, a_2, \ldots, a_n)$. These features could be informative about the characteristics of a node or edge; for instance, node features could contain review token counts, and edge features could indicate the type of edge that exists between two nodes (which could be more than 1). Any graph that has features (i.e., feature vectors) assigned to nodes and/or edges is referred to as an attributed graph. Many graph-based techniques, such as GNNs, rely on these attributes.

## 2.2 Multi-Relation Fraud Graph

In this section, the graph $\mathcal{G} = (V, E)$ is defined by the set of nodes $V$ and edges $E$. Each edge in $E$ represents the connections among nodes in fraud detection graph-based problem. Thereafter, we present how to apply RGCN to multi-relation fraud detection problems.

A multi-relation graph is defined as $\mathcal{G} = \{V, X, \{E_r\}|_{(r=1)}^{R}, Y\}$, where $V$ is the set of nodes $(v_1, \ldots, v_n)$. Each node $v_i$ has a d-dimensional feature vector $x_i \in R_d$ and $\boldsymbol{x} = \{x_1, \ldots, x_n\}$ represents a set of all node features. $e_{(i,j)}^{r} = (vi, vj) \in E_r$ is an edge between $v_i$ and $v_j$ with a relation $r \in \{1, \ldots, R\}$. Note that any edge can be associated by several relations, and there are $R$ different relation types. $Y$ is the label for each node in V.

## 2.3 Problem Statement

The target entity in the fraud detection problem is represented by node v, whose suspicions should be vindicated, which can be for example, viewed in a trading system transaction or a review on the reviews website. Label assigned to the node is $y_v \in \{0, 1\} \in Y$, where 0 denotes normal and 1 denote suspicious or fraudster. The relation might be communications, rules, or any common properties across nodes for example, two reviews from the same user or transactions from the same devices. Node classification problem is a graph-based fraud detection model that is trained using both the multi-relationship graph and the labeled node data. Then, the converged models are used to predict the fraud unlabeled nodes.

# 3 Background and Related Work

This section provides a concise summary of the background and pertinent research on graphs, Graph Neural Networks (GNNs), and anomaly detection broadly. It also discusses the application of random walks in graph anomaly detection.

## 3.1 Graph Neural Network

Graph Neural Network (GNN) is a class of deep learning models that was created especially to interpret graph-structured data and learn both attributed and structural graph information (Kipf & Welling, 2017), (Hamilton et al., 2017). GNN's primary function is to learn node representations by gathering and spreading data from nearby nodes to the center node. GNNs can capture intricate topology dependencies and contextual information among nodes through propagation, or message passing (Wu et al., 2021), (Bei et al., 2025).

There are two categories of GNNs: GNN spatial-based and GNN spectral-based. Spatial-based GNNs use the spatial information of the nodes and message-passing techniques Zhu et al. (2021) and (Hamilton et al., 2017). Spectral-based GNNs, on the other hand, use spectral graph theory and the graph's Laplacian matrix (Kipf & Welling, 2017), (Dong et al., 2024), and (He et al., 2024). Due to their ability to capture node properties and graph structure information, Graph Neural Networks (GNNs) have achieved reasonable success in various tasks (e.g., node classification, sub-graph classification, graph classification, and link prediction) (Yu et al., 2022b).

Another direction was mentioned in Yu et al. (2022a): there has already been some work done on modeling the heterogeneous graph representation. For instance, metapath2vec Dong et al. (2017) and HERec Shi et al. (2019) are two studies that use random walks to build meta-paths over the heterogeneous network for node embeddings. GNNs are becoming more and more popular as a way to encode methods for graph structures. Several heterogeneous GNN models have been developed to improve GNN architecture by enabling them to add the capacity of capturing heterogeneous contextual information at the node and edge levels. Multi-relation graphs represent a specific category of heterogeneous graph structures.

## 3.2 Anomaly Detection on Graph Data

Early anomaly detection methods used simple GNNs. GCN-based Anti-Spam (GAS) (Li et al., 2019) used predetermined aggregators over homogeneous similarity graphs, while FdGars Wang et al. (2019) used integrated preset user tagging with multi-relational GCN to detect fraudulent users (Yu et al., 2024). Recently, the Rayleigh Quotient GNN Dong et al. (2024) extracts spectral energy features to detect anomalous graph structures, while ADA-GAD He et al. (2024) employs a two-stage anomaly-denoised augmentation with score-distribution regularization to mitigate the influence of anomalous shapes on embedding learning.

Furthermore, Dou et al. (2020) delineate fraudster disguise along two dimensions: feature disguise, wherein fraudsters alter behavior, incorporate special characters (spamouflage), or employ language generation to circumvent feature-based detectors; and relational disguise, wherein fraudsters forge associations with high-credit legitimate users to mitigate suspicion within the graph.

Recent GNN-based fraud detection systems tackle four fundamental challengesnoise, camouflage, class imbalance, and heterophily each from a unique perspective. GraphConsis Liu et al. (2020) mitigates the influence of noisy neighbors by enforcing embedding consistency. CARE-GNN Dou et al. (2020) enhances this approach through a reinforcement learning method through selecting neighbors in multi-relational graphs, dynamically selecting salient neighbors to mitigate camouflage. PC-GNN Liu et al. (2021) partitions the graph into equitable substructures to provide adequate representation of minority fraudulent nodes. FRAUDRE Zhang et al. (2021) encapsulates various relational neighbor patterns and mitigates imbalance via a specialized loss function. RioGNN (Peng et al., 2021) separates relation-dependent representations to address heterophily in mixed-label neighborhoods. DOS-GNN (Jing et al., 2024) integrates dual-feature aggregation with embedding-space oversampling to address heterophily and imbalance concurrently; nonetheless, it is limited to single-relational contexts and has poor generalization to complicated relational graphs.

Multi-relational GNNs can capture complicated entity interactions across link types, making them a popular knowledge representation tool (Tian & Meng, 2024). Significantly, Dou (2022) illustrates that astute fraudsters manipulate this framework by linking with legitimate users to diminish suspicion, resulting in GNNs acquiring non-discriminative embeddings for concealed anomalies. Although these methods offer individual benefits, they either entail considerable computational expenses via reinforcement learning or focus solely on one aspect of fraud detection. None concurrently tackle camouflage, community-level collusion, heterophily, and class imbalance without depending on reinforcement learning or synthetic oversampling, which is the specific gap that RWR-RGCN addresses.

Table 1 delineates the principal architectural distinctions between RWR-RGCN and preceding methodologies across five categories of fraud detection. CARE-GNN (Dou et al., 2020) uses reinforcement learning to filter neighbors and alleviate camouflage; nevertheless, it performs solely at the node-to-node level and is incapable of identifying group-level collusion. PC-GNN (Liu et al., 2021) tackles class imbalance via balanced resampling; nevertheless, it lacks a community detection or disguise method. FRAUDRE (Zhang et al., 2021) identifies relational neighbor patterns and mitigates imbalance using a specialized loss function, partially tackling relational camouflage by embedding disentanglement; nevertheless, it does not incorporate community structure or address heterophily. RioGNN (Peng et al., 2021) attains a high AUC via RL-based relational disentanglement, although it overlooks community structure and necessitates expensive iterative RL training. DOS-GNN (Jing et al., 2024) addresses heterophily and imbalance via dual aggregation and embedding-space oversampling; nonetheless, it fails to simulate community structure and may introduce synthetic noise.

No current methodology concurrently tackles camouflage, community-level collusion, heterophily, and class imbalance without employing reinforcement learning or synthetic augmentation - a gap that RWR-RGCN bridges by merging RWR, Louvain, and RGCN as three interrelated components, each enhancing the input quality of the subsequent one. RWR-RGCN is based on the notion that concealment at the feature level, collusion at the community level, and heterophily at the relation level require three complimentary signals. RWR solves the first by producing individualized walk sequences for each node, which creates discriminative path-level characteristics that reveal fraudster and normal user behavioral variance even when node features are hidden. Louvain solves the second problem by partitioning the graph into high-modularity communities, which retain fraud rings as cohesive structural units that are identifiable even when intra-ring connections appear normal. Third, RGCN learns relation-specific transformation matrices to capture the interaction patterns across fraud signal relations.

Table 1: Comparison of fraud detection methods across key capabilities (✓ = fully addressed, ✗ = not addressed, ◗ = partially addressed).

| Method | Camouflage | Community | Heterophily | Imbalance | RL-free |
|---|---|---|---|---|---|
| CARE-GNN (Dou et al., 2020) | ✓ | ✗ | ◗ | ✗ | ✗ |
| PC-GNN (Liu et al., 2021) | ✗ | ✗ | ✗ | ✓ | ✓ |
| FRAUDRE (Zhang et al., 2021) | ◗ | ✗ | ✗ | ✓ | ✓ |
| RioGNN (Peng et al., 2021) | ◗ | ✗ | ✓ | ✗ | ✗ |
| DOS-GNN (Jing et al., 2024) | ✗ | ✗ | ✓ | ✓ | ✓ |
| **RWR-RGCN (ours)** | ✓ | ✓ | ✓ | ✓ | ✓ |

## 4 The RWR-RGCN Framework

Current GNN-based fraud detection systems tackle camouflage, class imbalance, relational inconsistency, and heterophily via selective aggregation, neighbor sampling, and embedding disentanglement. Nevertheless, despite these advancements, all such methodologies perform at the node or edge level, constraining their capacity to identify group-level collusion, the principal evasion tactic employed by organized fraudsters.

The primary deficiency is in community-level concealment. Techniques like GraphSAGE (Hamilton et al., 2017), CARE-GNN (Dou et al., 2020), and RioGNN (Peng et al., 2021) alleviate camouflage via selective neighbor aggregation; however, they work exclusively on a node-to-node basis. While they can diminish dubious individual connections, they fail to identify fraud rings that are only discernible when nodes are analyzed as a cohesive structure. Louvain community detection mitigates this issue by partitioning the network according to global modularity, thus preserving fraud rings as cohesive substructures despite the appearance of normal individual connections.

Figure 1 illustrates the overall design of RWR-RGCN. The framework performs in three consecutive steps. Initially, Random Walk with Restart produces distinctive node sequences for each relation type, encapsulating the behavioral difference between fraudulent and regular nodes at the path level. Secondly, Louvain community identification divides the graph into high-modularity communities that suggest potential fraud rings, facilitating aggregation at the group level instead of the level of individual neighborhood. The RWR walk characteristics and Louvain community assignments are combined to create enhanced node representations, which are processed through an embedding layer and aggregated prior to entering the RGCN stack. Third, stacked RGCN layers facilitate relation-aware feature propagation among all relation types, employing LeakyReLU activation, dropout regularization, and concluding with a Softmax layer for binary fraud classification.

This design delivers three distinct advantages. The Louvain community structure reveals collusive fraud networks that fraudsters cannot conceal through individual connection tweaks. RWR-based feature selection identifies local camouflage patterns that conventional neighborhood aggregation overlooks. Louvain's nearly linear scalability (Huang et al., 2025) guarantees the framework's efficiency on extensive real-world

graphs while yielding interpretable community assignments that could be directly examined as potential fraud groups. Collectively, these advancements render RWR-RGNN a resilient, scalable approach for fraud detection. The following subsections formally define the framework details and its different parts and explain the algorithm in detail.

## 4.1 The Proposed Framework Components

This subsection elucidates the rationale behind each step of the proposed fraud detection framework. The approach leverages three key components:

1. A depth-wise RWR-based feature selector that tracks the fraudster's evolving camouflage patterns,
2. A vertex-community-detection-based aggregator that summarizes node behavior within local subgraphs, and
3. A Relation Graph Convolutional Network (RGCN) for effective feature learning across diverse graph relationships. A primary challenge in graph-based fraud detection lies in generating informative features that reliably distinguish fraudulent nodes from legitimate ones. To address this challenge, a versatile graph processing framework capable of classifying nodes across various graph relationships is introduced.

Each of the components is presented in the following before introducing the exact algorithm steps, which utilize all three components to classify the nodes as either fraudsters or normal for an undirected graph.

### 4.1.1 Fraud Neighbors Selectors

RWR Jin et al. (2019) calculates each node's proximity to a specified query node $s$ in a graph. It is often referred to as Personalized PageRank (PPR) with a single seed node. The random surfer assumed by RWR begins at node $s$. With a likelihood of $1 - c$, the surfer goes to one of its nearby nodes, or with a probability of $c$ it restarts at node $s$. Every neighbor $v$ that the surfer goes from $u$ to its neighbor is chosen with a probability that is proportionate to the weight in the edge $(u, v)$. A high score indicates that the nodes $s$ and $u$ are highly connected.

Since RWR offers a personalized ranking with respect to a node Jin et al. (2019), it has been utilized in numerous graph applications, including detecting community, graph partitioning, graph matching, graph sampling Nakajima & Shudo (2022), link prediction, and ranking. RWR ranking findings were utilized by Sun et al. (2005) to identify anomalies. Empirical research by Gleich & Seshadhri (2012) has demonstrated that random walk-based models are competitive with other cut-based methods for identifying graphs of local clustering. Our assumption when using RWR is to find the fraud versus normal paths within each nodes neighborhood. Through experimentation, we found that the features generated from the $i$-th path were more discriminative to classify each node as either fraud or normal.

### 4.1.2 Vertex Community Detection Based Aggregator

Fraudsters behaving in structured groups typically exhibit intense interactions among themselves while sustaining few linkages to the wider network, leading to high-modularity communities. Conventional aggregation techniques at the node-level fail to identify this collaboration as they assess each node's adjacent neighbors in isolation, overlooking the collective signal that reveals orchestrated fraud. We employ Louvain community detection (Blondel et al., 2008) to partition the graph into coherent structural communities, maintaining entire fraud rings like unified components despite particular intra-ring connections appearing normal.

In 2 Louvain enhances modularity, which is a scalar metric of the density of intra-community connections compared to inter-community connections, constrained between -1 and 1 (Blondel et al., 2008) via two iteratively repeated phases. Initially, each node resides within its own distinct community. For each node $i$, the program assesses the modularity gain of relocating $i$ to each adjacent community. Node $i$ joins the community that provides the highest positive benefit; if no positive benefit is available, it remains in its existing community. This procedure iterates systematically across all nodes until no individual adjustment enhances modularity, achieving a local maximum. During the second phase, all nodes within each identified community are consolidated into a singular super-node, resulting in a new coarsened graph, after which the

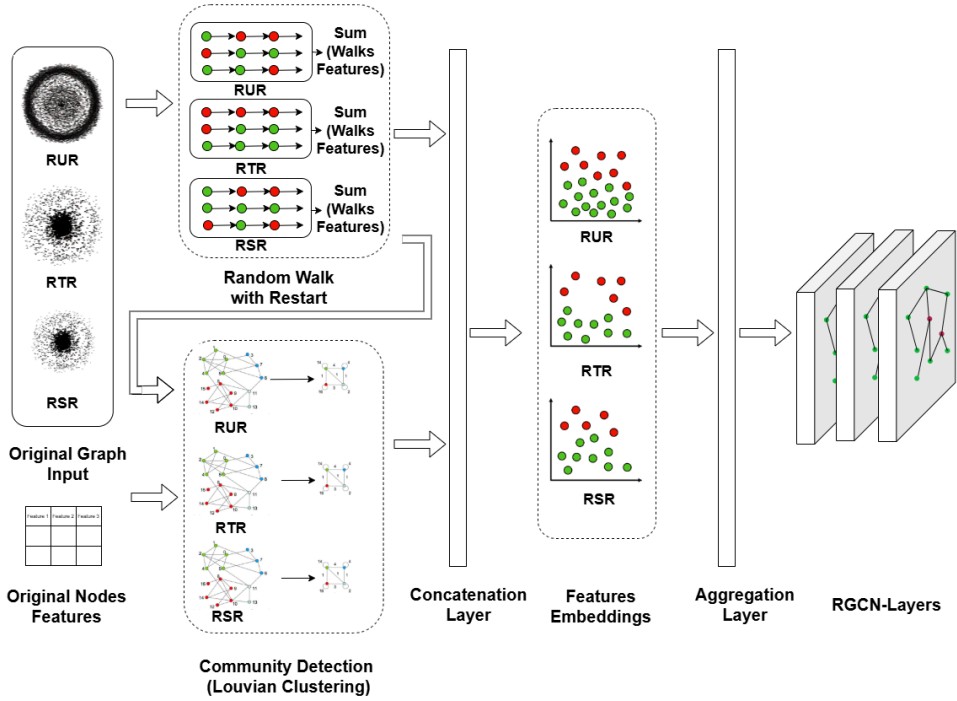

(a) The initial phase of the RWR-RGCN architecture commences with the input layer, containing the original relational graph data and the node features linked to each entity (for further details, see Section 4.1 Dataset). This input undergoes processing via the Random Walk with Restart (RWR) for each relation type, generating walk-based feature representations. The original graph and RWR outputs are subsequently input into the community discovery module, employing Louvain clustering to ascertain structural communities. The outputs are concatenated and input into the feature embedding layer, followed by an aggregation phase that merges the learned representations before they are processed through the stacked RGCN layers for additional relational reasoning illustrated in part (b).

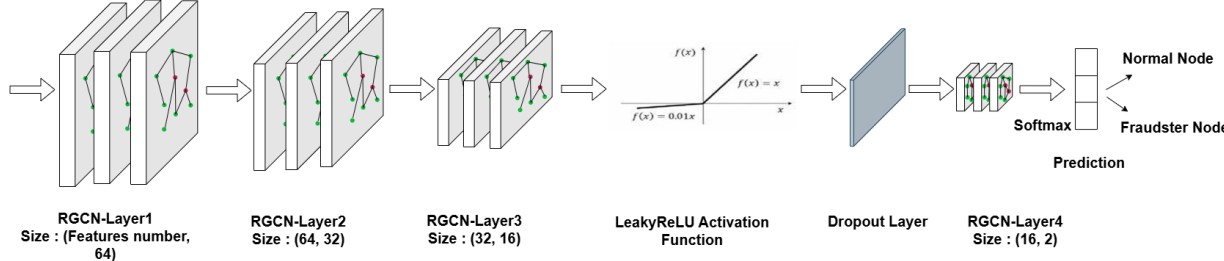

(b) The output from (a) will be inputted to three consecutive RGCN layers that systematically diminish feature dimensionality: the first layer converts input data into 64-dimensional representations, the second compresses them to 32 dimensions, and the third further reduces them to 16 dimensions. A LeakyReLU activation function is utilized to introduce non-linearity, while a dropout layer is employed to mitigate overfitting. The processed characteristics are subsequently transmitted to a final RGCN layer, which transforms the 16-dimensional representations into a 2-dimensional output space. A softmax function is utilized on these outputs to produce a prediction, classifying each node as either Normal or Fraudster.

Figure 1: A high level architecture proposed applying in node classification methods for multi-relation graph data. It consists of two parts (a) and (b) (see Section 4.1 for details)

first phase is reapplied. The two phases alternate until modularity stabilizes and no additional enhancement can be achieved.

Within the RWR-GNN framework, the community assignment generated by Louvain is merged with the RWR walk features to create the enhanced node representation inputted into the RGCN layers. This enables the model to incorporate insights from both the local behavioral signal obtained by RWR and the global structural signal derived from community membership, thereby revealing fraudsters who may conceal their individual actions but cannot obscure their collusive group dynamics.

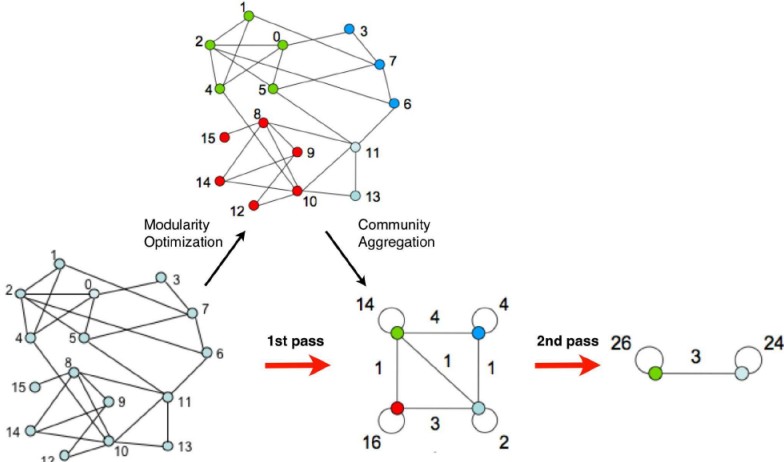

Figure 2: Blondel et al. (2008) visualizes the phases in order to optimize the modularity for louvian clustering model

### 4.1.3 Relational Graph Convolution Network (RGCN) As Node Classifier

RGCN is defined from labeled multi-graphs as $\mathcal{G} = (V; E; R)$ with nodes (entities) $v_i \in V$ and labeled edges (relations) $(v_i; r; v_j) \in E$, where $r \in R$ is a relation type. In Schlichtkrull et al. (2018) the authors innovate RGCN as a GCN extension to find the neighborhoods in local graph but for multi-relation data. We consider a node classification task to classify each node (review) in the graph fraud as or not and analyze the neighbors from each relation of the same node.

R-GCN layers are used for (semi-)supervised classification of nodes (entities) and a softmax layer (per node) on the final output layer. On every labeled node, we minimize the cross-entropy loss and then predict the test set for unseen data.

$$\mathcal{L} = -\sum_{i \in \mathcal{Y}} \sum_{k=1}^{K} t_{ik} \ln h_{ik}^{(L)} \tag{1}$$

In (Eq. 1) where $\ln h_{ik}^{(L)}$ is the $k$-th element of the model output for the $i$-th labeled node, and $y$ is the set of node actual labels. The associated ground truth label is shown by $t_{ik}$. In the experiments, we use (full-batch) gradient descent methods to train the model.

### 4.1.4 Algorithm Details

Algorithm 1 presents the pseudocode for the proposed RWR-RGCN framework. The algorithm takes as input a multi-relation undirected graph, its edge index, a query node, random walk parameters (path length and restart probability), initial community assignments, a new community label, and node features. Initialization

includes a path to store node sequences from random walks, edge weights ($w$), a restart vector ($s$) initialized with a single 1 at the query node, a row-normalized adjacency matrix $\boldsymbol{P}$, an iteration vector ($\boldsymbol{x}$) initialized to $s$, an error tolerance ($e$) for power iteration, and the initial community assignments ($C$). The RWR-RGCN algorithm begins by establishing a heterogeneous, multi-relational network, initialized with the nodes' original feature sets. Subsequently, it employs Random Walks with Restart (RWR) to trace potential fraud and normal pathways, using pre-defined threshold sequences. The restart probability, alpha ($\alpha$), dictates the scope of these walks, favoring local structures with higher values. Next, the algorithm constructs enriched feature vectors by aggregating node features from these identified pathways. Then, community structures are detected using the Louvain clustering technique, which iteratively optimizes modularity until stable communities are formed. Finally, the augmented node features and community assignments are passed into a Relational Graph Convolutional Network (RGCN), where, for each layer or relation, the RGCN obtains unique feature embeddings for the existing nodes within that layer and calculates message passing to aggregate information from neighboring nodes; this processed information is then passed to a HeteroRGCN layer with a LeakyReLU activation function, which integrates information from all relations to produce logits that determine the class label (fraud or normal) for each node.

## 5 Performance Evaluation

Due to the inherent class imbalance in fraud detection, specialized evaluation metrics are crucial, as traditional accuracy can be misleading. To accurately assess model performance, we focus on recall, which measures the model's ability to identify all actual fraudulent instances, a critical metric in imbalanced scenarios where minimizing false negatives is paramount, and Receiver Operating Characteristic Area Under Curve (ROC-AUC), which quantifies the model's performance across various decision thresholds by depicting the trade-off between true positive and false positive rates. A high ROC-AUC indicates robust performance, demonstrating the model's ability to discriminate between fraudulent and legitimate activities regardless of the chosen threshold. As pointed out in (Sun et al., 2005), these metrics provide a comprehensive and objective evaluation of fraud detection models, particularly in the presence of significant class imbalance. And to ensure meaningful comparison with prior work, the evaluation has been conducted on the Yelp dataset, which is commonly used in the literature for benchmarking fraud detection models.

The comparative experiments have been performed between RWR and eight different baseline models and the latest state-of-the-art models, which were specifically developed for fraud detection or imbalanced graphs, which are briefly introduced as follows:

- GCN (Schlichtkrull et al., 2018): is a representation of the vanilla graph convolution approach, establishing a straightforward and well-structured hierarchical propagation rule for neural network models.
- GraphSAGE (Hamilton et al., 2017): is a representative non-spectrogram approach. This method offers a comprehensive inductive framework for each node, which samples and aggregates the features of its local neighbors to produce an embedding, rather than training an independent embedding. It enhances the scalability and adaptability of GNNs.
- GraphConsis (Liu et al., 2020): is a model that integrates context embedding with nodes, eliminates inconsistent neighbors, and produces related sample probabilities. The embeddings of sampled nodes from each relation are integrated utilizing a relation attention technique.
- CARE-GNN (Dou et al., 2020): a layer employing a label-perceived similarity metric is utilized to identify information-rich nearby nodes. Uses a special approach to select the most informative neighboring nodes for aggregation
- PC-GNN (Liu et al., 2021): uses samplers to build subgraphs and sample informative neighbors for aggregation.
- FRAUDRE (Zhang et al., 2021): a model that separates relation-specific embeddings and reconstructs them to improve resilience against camouflage and relationship inconsistencies in multi-relational graphs.
- RioGNN (Peng et al., 2021): a Label-aware similarity measure employs a two-layer framework for neighbor selection. Employs the Actor-Critic (AC) algorithm with a discrete strategy to itera-

---

**Algorithm 1** RWR-Relation-GCN

---

1: **Inputs:** Edge index $e$, starting node $v_0$, path length $t$, restart probability $\alpha$, graph $\mathcal{G}$, community $C$, new community $new\_C$, feature input $\boldsymbol{f}$, layer $L$

2: **Output:** Logit for each node $v_0 \in (0,1)$

3: **Initialize:** $path = [v_0]$ (path stores nodes in a walk), $w$ (weight from $v_i$ to $v_{i+1}$), $s$ (restart vector with all entries 0 except 1 at $v_0$), $\mathbf{P}$ (row-normalized adjacency matrix), $x := s$, $e$ (error tolerance for power iteration), $C$ (initial communities)

4: **for** $v_0 = 1$ to $V_n$ **do**

5:     **for** *path length* $<= t - 1$ and $v_i =! v_n$ **do**

6:         **while** $x$ has not converged **do**

7:             $x := \alpha_s + (1+\alpha)P_x{}^T$

8:             $w = w(e(v_i, v_{i+1})); w \in (0;1)$

9:             **if** $w$ is the maximum **then**

10:                 $rwr\_score = x$

11:                 return $rwr\_score$, $path+ = v_i$, $v_i + 1$

12:             **else**

13:                 restart

14:             **end if**

15:         **end while**

16:     **end for**

17: **end for**

18: **while** $G.nodes == length(G.nodes)$ **do**

19:     $C = v0, j$

20:     $N = $ neighbors to each $v_{0,j}$

21:     $C = C.append(N)$

22:     $sum(modularity(N)); modularity \in (0;1)$

23:     **if** Modularity After $N >=$ Modularity before $N$ **then**

24:         Add node $v_i$ to $new\_C$

25:     **else**

26:         $v_i \in C$

27:         return $new\_C$

28:     **end if**

29: **end while**

30: **for** $n$ in *path* with max($rwr\_score$) **do**:

31:     $\boldsymbol{f}(V) = sum(F$ for each $n)$

32:     $\boldsymbol{f}(V) = Append(new\_C(v_i))$

33: **end for**

34: **for** $L = 1$ to $|\boldsymbol{f}_i|$ **do**

35:     $e_f^{(l)} \leftarrow \text{Embed}(\boldsymbol{f}, l)$ for all $\boldsymbol{f} \in \mathbf{F}, l \in L$

36:     $Wh = e_f^{(l)}$

37:     Calculate $h$ message passing

38:     $h = Wh * h$

39: **end for**

40: **while** not convergence **do**

41:     $HeteroRGCNLayer(G, h)$

42:     $LeakyReLU$

43:     return $logit(n)$

44: **end while**

---

tively determine the filter thresholds for various relationships, utilizing these thresholds as relational weights to aggregate neighboring entities across multiple relations.

- BWGNN (Tang et al., 2022) a spectral method employing Beta Wavelet filters to effectively determine high-frequency anomaly signals through band-pass filtering.
- DOS-GNN (Jing et al., 2024): a dual-feature aggregation framework that maintains both similarity and dissimilarity signals while implementing oversampling in the embedding space to mitigate class imbalance in fraud detection.
- DGA-GNN (Duan et al., 2024) a spatial approach utilizing decision tree binning encoding and feedback dynamic grouping for neighbor hierarchical aggregation.
- LEX-GNN (Hyun et al., 2024) a framework for label exploration that distinguishes between message passing and reception according to predicted fraud probabilities.

## 5.1 Dataset

The proposed framework is evaluated the node classification task using the selected benchmark datasets, Yelp and Amazon. Yelp is review datasets focus on hotels and restaurants, which filtered the reviews as spam and recommended them as normal reviews. The Amazon dataset consists of musical instruments product reviews. Identify users with more than 80% advantageous votes as benign and those with less than 20% as fraud. And the datasets have been used to study fraud detection in multi-relation graph convolution networks (Dou et al., 2020). Yelp includes 32 handcrafted features, while Amazon includes 25 features only.

Dou et al. (2020) datasets [1] designate their datasets as multi-relation graphs, with Yelp consisting of three relation types. 1) R-S-R: It links reviews that belong to the same product that has the same star rating (15); 2) R-U-R: It links reviews written by the same user; 3) R-T-R: It links two reviews that were posted in the same month and belong to the same product. The Yelp dataset contains 45,954 nodes with 14.5% of fraud nodes, meaning around 6,663 nodes are fraudsters with a total of 3,846,979 edges. It divided the relations into 49,315 edges, 573,616 edges, and 3,402,743 edges for R-U-R, R-T-R, and R-S-R, respectively.

Amazon contains users as nodes within the graph and establishes three relationships: 1) U-P-U: linking users who have reviewed at least one identical product; 2) U-S-V: connecting users who share at least one identical star rating within a week; 3) U-V-U: associating users with the top 5% of mutual review text similarities, quantified by TF-IDF, among all users. The Amazon dataset consists of 11,944 nodes with 9.5% of fraud nodes, meaning around 1,134 nodes are fraudsters with a total of 4,398,392 edges. The division of relations is 175,608 edges, 3,566,479 edges, and 1,036,737 edges for U-P-U, U-S-U, and U-V-U, respectively

## 5.2 Evaluation Metrics

In most classification tasks, the methods used to evaluate the models proposed are confusion matrices that can summarize the results and errors in the training pipeline, table 2 shows that all performance matrices match.

Table 2: Confusion Matrix

|  | **Actual positive** | **Actual negative** |
| --- | --- | --- |
| **Predicted positive** | True positive (TP) | False positive (FP) |
| **Predicted negative** | False negative (FN) | True negative (TN) |

Gleich & Seshadhri (2012) mentioned that there are many matrices to evaluate the proposed model performance, and we used AUC and recall as popular assessments for binary classification.

Recall in (Eq. 2), or the True Positive Rate (TPR), calculates the ratio of the positive class that the model accurately anticipated to be positive. Unbalance has no effect on recall since it solely depends on the positive class. The number of negative samples that are incorrectly identified as positive is not taken into account

---

[1]Yelp and Amazon datasets link: https://github.com/YingtongDou/CARE-GNN/tree/master/data

by recall, which can be problematic in situations where there is a class imbalance in the data and a large number of negative samples.

$$\text{Recall} = TPR = \frac{\text{TP}}{\text{TP} + \text{FN}} \tag{2}$$

The receiver operating characteristics (ROC) curve is a method for evaluation that plots the false positive rate over the true positive rate, putting together a visual representation of the trade-off between accurately and wrongly labeled positive and negative data points. Gleich & Seshadhri (2012) Thresholding can be employed in models that generate continuous probabilities to construct a series of points along ROC space. Using this, a solitary summary measure is the area under the ROC curve (AUC), which is frequently used to evaluate the effectiveness of different models.

$$\text{G-Mean} = \sqrt{\text{TPR} \cdot \text{TNR}} = \sqrt{\frac{\text{TP}}{\text{TP} + \text{FN}} \cdot \frac{\text{TN}}{\text{TN} + \text{FP}}} \tag{3}$$

G-Mean (Eq. 3), mean of True Positive Rate (TPR) and True Negative Rate (TNR) instantaneously retaining both values relatively balanced. The higher scores of the G-Mean indicate a high performance of the approaches (Shaha et al., 2021).

## 5.3 Evaluation Setup

This section displays the results from binary node classification task. Additionally, the proposed framework is tested in different random walks in figure 3. In the experiments, we used Yelp and Amazon multi-relations undirected graph datasets for a review system. In which Yelp has 2 classes of nodes: 39,291 normal and 6,663 fraudsters, with 14.5% of fraud nodes. The Yelp dataset was split into three divisions: 40%, 30%, and 30% for training, validation, and testing, respectively.

Table 3: Presents the performance of proposed approach RWR-RGCN on the Yelp dataset (%) though the experiments for number of walks.

| #Walks | AUC | Recall | G-Mean |
|--------|-------|--------|--------|
| 2 | 50.04 | 50.00 | 50.00 |
| 3 | 51.25 | 50.18 | 50.00 |
| 4 | 50.02 | 50.00 | 50.00 |
| 5 | **82.58** | 92.46 | **75.19** |
| 6 | 82.36 | **94.56** | 74.89 |
| 7 | 81.79 | 92.61 | 74.77 |
| 8 | 82.09 | 91.06 | 75.24 |
| 9 | 69.52 | 79.83 | 68.67 |

The results of fraud detection are visible in table 3, which uses test data from the Yelp multi-relation graph to identify the fraud nodes. All the models are perform on random walks from 2 to 9. The performance of proposed system was assessed and evaluated using AUC, recall, and G-Mean results, and the recall are demonstrated in figure 3 charts, highlighting the performance matrix across the epochs till models are converged. With the aim of calculating the performance matrices, we first calculated True Positive (TP), False Positive (FP), False Negative (FN), and True Negative (TN) for the dataset Yelp.

Figure 3 shows that the results are different in each random walk. It is fascinating to inspect when random walks are getting longer; the RGCN results become different. The result with random walks from 5 to 8 was the best. Such behavior is expected since the random walk captures and selects the fraud features from the neighbors node, which differentiates between normal and abnormal behavior in the representation learning. In contrast with the result in random walks 2 to 4 and 9, the worst is 9, as the small random walk has a few features to represent the differentiable behavior to be captured, or the higher random walk is the distraction

for differentiable properties of the features deteriorating after 9 random walks and more, leading to reduced smoothness or continuity in capturing the fraud structure and properties.

All factors considered, these outcomes demonstrate the strength of RGCN layers and proposed implementation. Since the fraud selector and two aggregation layers, one from the community detection and one from RGCN across different relations, are made to share the features of nodes in the convolution process. These benefits are vital in fraud detection applications.

## 5.4 Evaluation Results

The comparison between our proposed RWR-RGNN and a wide range of baseline methods has been conducted to fully evaluate its effectiveness. These include standard GNNs like the Graph Convolutional Network (GCN) Schlichtkrull et al. (2018) and GraphSAGE (Hamilton et al., 2017), as well as more advanced graph-based techniques for detecting fraud and anomalies that are specifically designed for spatial heterophily, such as GraphConsis (Liu et al., 2020), CARE-GNN (Dou et al., 2020), PC-GNN (Liu et al., 2021), FRAUDRE (Zhang et al., 2021), RioGNN (Peng et al., 2021), DGA-GNN (Duan et al., 2024), LEX-GNN (Hyun et al., 2024), and the latest DOS-GNN (Jing et al., 2024) also the spectral GNN such as BWGNN (Tang et al., 2022). These models represent the progress in mitigating noise, camouflage, imbalance, and heterophily in fraud detection.

GraphConsis Liu et al. (2020) specifically addresses inconsistencies in context, features, and relationships, which make it more robust in different situations. CARE-GNN Dou et al. (2020) is a model that is resistant to camouflage and uses reinforcement learning to adaptively sample neighbors. This technique solves the problem of fraudsters pretending to be real nodes. PC-GNN Liu et al. (2021) uses a two-step "pick and choose" method to resample neighbors and keep a balanced label distribution around nodes that are fraudulent. RioGNN Peng et al. (2021) uses relation-aware message passing with disentanglement to effectively capture heterophilic connections. FRAUDRE Zhang et al. (2021) propounds this concept by disentangling relation-specific embeddings and reconstructing them to improve resilience against relational inconsistency and complication. It is especially efficacious for multi-relational fraud graphs, wherein interactions encompass people, goods, and transactions. DOS-GNN Jing et al. (2024) introduces a dual-feature aggregation framework that clearly preserves both similarity and dissimilarity signals in embeddings, utilizing oversampling in the embedding space to address class imbalance.

The experimental results emphasize the progression from conventional GNNs to innovative fraud detection frameworks in Yelp dataset and illustrates the efficacy of proposed RWR-RGCN. Baseline models like GCN Schlichtkrull et al. (2018), and GraphSAGE Hamilton et al. (2017) exhibit constrained performance, attaining AUCs of 53.38 and 54.39, respectively. Their dependence on local neighborhood aggregation and homophily assumptions undermines their efficacy in fraud detection, since fraudsters deliberately link to normal nodes, leading to suboptimal G-Mean values (47.36 for GCN Schlichtkrull et al. (2018) and merely 25.89 for GraphSAGE (Hamilton et al., 2017).

Modern specialized fraud detection models exhibit markedly improved performance by tackling specific difficulties mentioned in table 4. In Yelp dataset GraphConsis Liu et al. (2020) enhances consistency among relations and features, resulting in an increased AUC of 69.55 and recall of 66.20. CARE-GNN Dou et al. (2020) progresses by using reinforcement learning for neighbor filtering, attaining an AUC of 75.70 and a G-Mean of 67.91. PC-GNN Liu et al. (2021) and FRAUDRE Zhang et al. (2021) exhibit competitive performance as well. PC-GNN Liu et al. (2021) equilibrates label distributions via resampling and achieves the greatest G-Mean of 70.88 among these models, whereas FRAUDRE Zhang et al. (2021) employs relational disentanglement to attain a balanced AUC of 72.22 and a G-Mean of 69.78. RioGNN Peng et al. (2021), which improves relation-aware message forwarding, achieves the greatest AUC among current baselines at 82.38, demonstrating the benefits of simulating multi-relational fraud interactions. DOS-GNN (Jing et al., 2024), employing dual-feature aggregation and an oversampling method, attains a well-balanced performance (AUC 81.15, Recall 82.14, G-Mean 81.66), establishing it as a formidable contender, especially in addressing imbalanced fraud situations.

Our model exhibits consistently enhanced recall relative to BWGNN (Tang et al., 2022), DGA-GNN (Duan et al., 2024), and LEX-GNN (Hyun et al., 2024), attaining improvements of 37.87%, 10.33%, and 8.21%

respectively in Yelp dataset. This improvement is primarily attributed to the integration of Random Walk with Restart, community-aware clustering, and relation-specific aggregation, which collectively augment the model's capacity to detect varied and subtly expressed fraud indicators. Unlike BWGNNs (Tang et al., 2022) dependence on band-pass spectral Beta wavelet filtering, which may overlook anomalies not confined to particular frequency ranges, our methodology investigates wider structural contexts, enhancing the detection of sparsely linked fraudulent nodes using model community-level collusion structure. In contrast to DGA-GNN (Duan et al., 2024), addresses non-additive features and grouped message distinguishability by decision tree binning and dynamic grouping, emphasizing feature encoding and neighbor grouping at the node level rather than maintaining the integrity of the ring structure through community detection which may restrict sensitivity to nuanced patterns due to division in grouping, our community-guided representation offers more consistent group-level signals. In contrast to LEX-GNN (Hyun et al., 2024), which bases message passing on predicted labels and may transmit early prediction inaccuracies, our relation-aware aggregation maintains minority fraud signals throughout the propagation process.

The results shown in Table 4 indicate that the suggested RWR-RGCN method exhibits competitive albeit inconsistent performance relative to baseline alternatives. Upon analyzing the AUC and Recall metrics on Amazon, the methodology demonstrates more moderate outcomes. With 5 walks, it attains an AUC of 78.76% and a Recall of 55.70%, however with 6 walks, these metrics improve somewhat to 79.32% and 57.54%, respectively. The scores are significantly inferior to the top baseline approaches such as BWGNN (97.42% AUC, 85.87% Recall) and DGA-GNN (98.39% AUC, 87.50% Recall). The G-Mean metric demonstrates comparable performance at 68.55% and 70.35% for the two walking configurations, which falls short of leading performers such as CARE-GNN (85.62%) and PC-GNN (87.82%).

Table 4: Experimental results (%) for node classification of different fraud detection methods on benchmark dataset. Some GNN models are highlighted bold denotes a significant improvement of results on the Yelp dataset.

| Group | Method | Yelp | | | Amazon | | |
|---|---|---|---|---|---|---|---|
| | | AUC | Recall | G-Mean | AUC | Recall | G-Mean |
| **Baselines** | GraphSAGE Hamilton et al. (2017) | 54.39 | 50.00 | 25.89 | 71.49 | 50.00 | 54.49 |
| | GCN Schlichtkrull et al. (2018) | 53.38 | 50.43 | 47.36 | 74.34 | 51.45 | 52.18 |
| | GraphConsis Liu et al. (2020) | 69.55 | 66.20 | 58.37 | 75.12 | 87.41 | 68.77 |
| | CARE-GNN (Dou et al., 2020) | 75.70 | 71.92 | 67.91 | 86.39 | 83.90 | 85.62 |
| | PC-GNN Liu et al. (2021) | 78.50 | 67.21 | 70.88 | 89.56 | 95.86 | 87.82 |
| | FRAUDRE Zhang et al. (2021) | 72.22 | 66.98 | 69.78 | 88.18 | 88.61 | **89.15** |
| | RioGNN Peng et al. (2021) | 82.38 | 75.08 | – | 96.19 | 89.82 | – |
| | BWGNN (Tang et al., 2022) | 90.54 | 56.69 | – | 97.42 | 85.87 | – |
| | DOS-GNN Jing et al. (2024) | 81.15 | 82.14 | **81.66** | 90.15 | 89.75 | 88.6 |
| | DGA-GNN (Duan et al., 2024) | **97.95** | 84.23 | – | **98.39** | 87.50 | – |
| | LEX-GNN (Hyun et al., 2024) | 96.40 | 86.35 | – | 97.91 | **93.48** | – |
| **Ours** | RWR-RGCN (#walks=5) | 82.58 | 92.46 | 75.19 | 78.76 | 55.70 | 68.55 |
| | RWR-RGCN (#walks=6) | 82.36 | **94.56** | 74.89 | 79.32 | 57.54 | 70.35 |

The suggested RWR-GCN surpasses or equals these leading baselines or state-of-the-art models across multiple criteria. Five random walks yield an AUC of 82.58, somewhat exceeding RioGNN Peng et al. (2021) and DOS-GNN (Jing et al., 2024), while significantly enhancing recall to 92.46. This model demonstrates its capacity to identify fraudulent nodes more thoroughly, which is essential in fraud detection applications where overlooking fraud incurs significant costs. The inclusion of six random walks elevates recall to 94.56, thus validating the efficacy of random-walk-based neighbor refinement in identifying fraudulent behavioral patterns. RWR-GCN balances representation learning quality and fraud sensitivity better than BWGNN (Tang et al., 2022), which uses spectral filtering to address class imbalance, DGA-GNN (Duan et al., 2024),

which uses adaptive graph augmentation, and LEX-GNN (Hyun et al., 2024), which uses label expansion strategies. Although RWR-GCN has slightly lower G-Mean values (75.19 for five walks and 74.89 for six walks) than DOS-GNN (81.66), the higher recall is often more important in real-world fraud detection scenarios, where minimizing false negatives is more important than marginal gains in balance metrics. While AUC can be misleading in highly imbalanced fraud contexts, Recall directly assesses fraud case capture, making it more meaningful. Yelp supports the preference for models with somewhat lower AUC but significantly higher recall.

## 5.5 Ablation Study

To assess the individual contribution of each component, we performed a systematic ablation study by sequentially deleting one component at a time from the complete RWR-RGCN framework tested on the Yelp dataset. (1) the complete model, (2) excluding Louvain (Louvain), where community features are disregarded and only RWR outputs are aggregated from three relations prior to RGCN; (3) excluding RWR (RWR), where original node features are directly input into Louvain + RGCN; (4) the RGCN backbone in isolation, utilizing solely original node features; and (5) the GCN devoid of relational processing.

The results in Table 5 indicates that the removal of Louvain clustering output in a modest yet significant decrease in recall (81.38%), suggesting that community-level features mitigate false positives by grounding node representations within a group-level structural framework. While the elimination of RWR leads to the most significant decline in recall (70.38%), so affirming that selecting neighbor using walk-based approach is the principal factor influencing the model's capacity to detect disguised fraudsters. The RGCN backbone, in isolation and devoid of either component, exhibits significantly diminished performance across all measures, indicating that relation-aware convolution alone is inadequate and that both preprocessing components are essential. Each component thus provides a unique and supplementary function to the overall system.

Table 5: Ablation study results (%) on the Yelp dataset showing the contribution of each component to overall fraud detection performance and improvement percentage from baseline GCN.

| Model Variant | RGCN | RWR | Louvain | AUC | Recall | Improvement % |
|---|---|---|---|---|---|---|
| (1) RWR-RGCN Full Model | ✓ | ✓ | ✓ | 82.36 | 94.56 | 44.13 |
| (2) RWR + RGCN | ✓ | ✓ | ✗ | 70.51 | 81.38 | 30.95 |
| (3) Louvain + RGCN | ✓ | ✗ | ✓ | 70.34 | 70.38 | 19.95 |
| (4) RGCN only | ✓ | ✗ | ✗ | 63.98 | 66.22 | 15.79 |
| (5) GCN | ✗ | ✗ | ✗ | 53.38 | 50.43 | — |

# 6 Discussion

In this work, we articulated a fair evaluation experimental setup for RGCN and accomplished extensive experiments on graph classification. Since the data used in the experiment for a multi-relation graph is a Yelp dataset, the RWR sequences for each node were the best choice for selecting the suggested nodes to extract the most informative features from neighbors, which made a good performance to highlight the disguise sequence or path that the fraudsters pattern follows. And its results outperform those of comparing with baseline models in Yelp dataset.

Overall, these results indicate that RWR-GCN proposals are an effective alternative that unifies sequence-level exploration with community and relation-aware aggregation. By relying on probabilistic neighbor weighting rather than reinforcement learning or oversampling, it avoids excessive computational complexity and synthetic distortion. The higher recall rates indicate its ability to reduce false negatives, while the high AUC values demonstrate balanced detection effectiveness. RWR-GCN is a robust framework that tackles the issues associated with node, relation, and oversampling-based models, facilitating comprehension and scalability for extensive fraud detection.

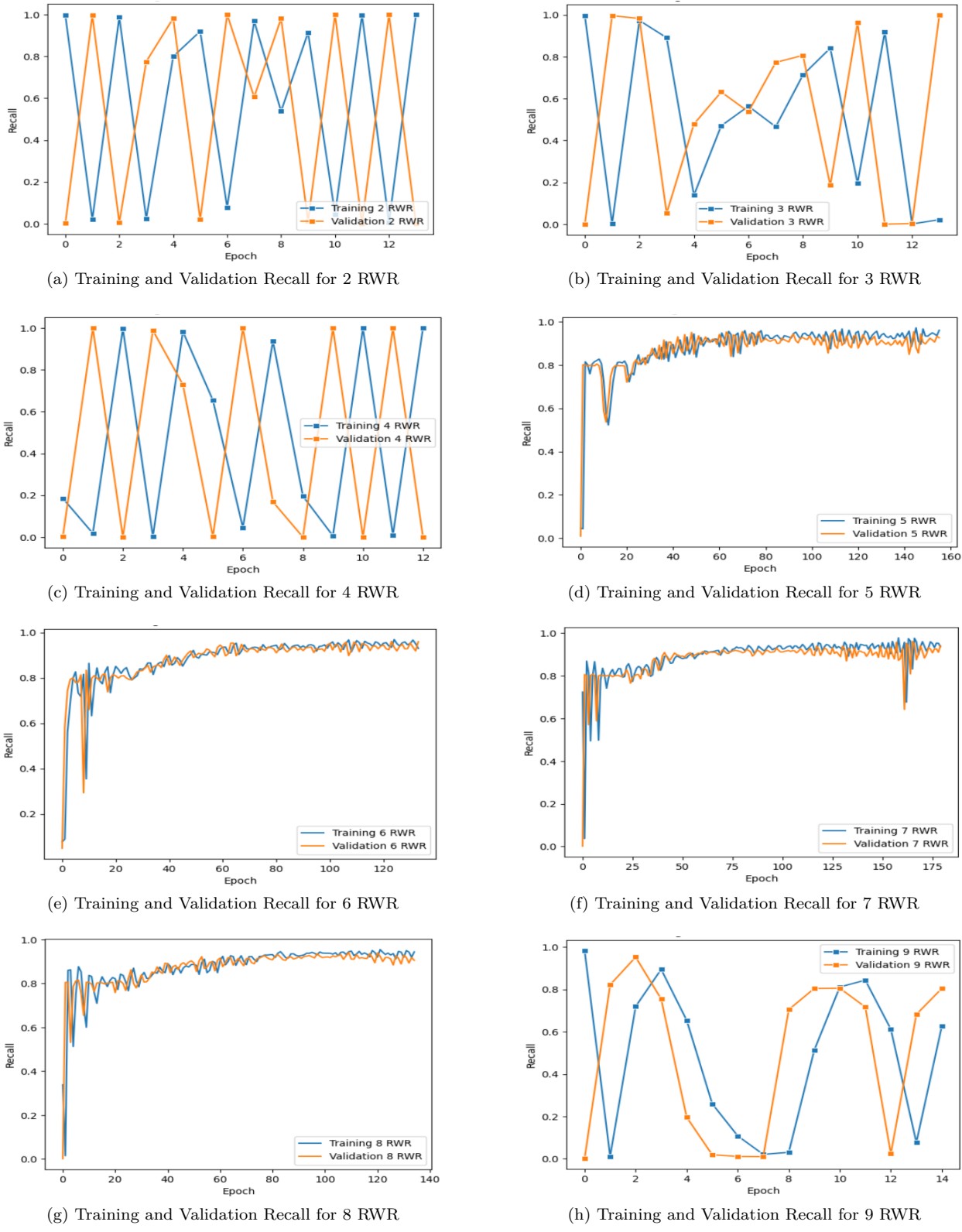

Figure 3: Recall performance matrix for different random walks from 2 to 9 walks starting from central nodes to select features as parameters in multi-relation node classification on both training and validation datasets

Lastly, it is interesting to see in figure 3 the relatively weak performance of the random walks between 2 to 4 and 9 walks. A possible explanation could be that the explicitly weak informative selection neighbors in the graph from each relation as long as the walks select informative sequences of nodes that provide features to the disguise behavior of the fraudster nodes. The extra aggregation layer that was added from the community detection and random walk models showed great improvements for multi-relation graph node classification. This will allow future work to concentrate on the node selection of fraud neighbors for better enhancement features of each node.

This draws attention to the innovative RGCN approach's shortcomings and raises several questions for further research and future work. Fraudsters who tend to disguise their deceptive behaviors by forming genuine connections with regular users pose a challenge to GNN. The suggested modifications to the local community model and random walks with restart, which have already been utilized as a model-based graph topology, produced significantly superior outcomes. For arbitrary node classification, the resulting model can serve as a more robust baseline. In addition, add an auto-encoder or node embedding layer to the architectures, which might enhance the model results. These developments should significantly help future node classification benchmarks.

A major strength of the proposed RWR-RGCN model is its ability to see through a fraudster's camouflage. Older methods often fail at this because they simply mix all of a user's connections together; if a fraudster hides a few fake reviews among hundreds of normal, everyday purchases, older models just average it all out and get completely fooled by the benign noise. The proposed method resists this tactic through two integrated mechanisms. First, instead of mixing everything into one generic pool, the relational component (RGCN) separates different types of actions, allowing the model to ignore the "normal" camouflage connections on the specific channels where malicious coordination is actually happening. Second, the restart mechanism (RWR) requires the algorithm to consistently revert to the target node during its exploration. This prevents the model from wandering off into the massive crowd of benign users the fraudster is using as a shield, ensuring it stays firmly anchored to the localized anomaly signal. However, it is important to note that if a fraudster's camouflage is so extensive that almost all of their relational channels appear completely normal; which is the fundamental challenge of the Amazon dataset, this method is still expected to struggle, as there are simply no strong malicious signals left for the architecture to isolate.

The proposed methods stark difference in performance between the two datasets is fundamentally tied to how its core mechanism-neighborhood aggregation-interacts with the structural clustering of the two platforms. On Yelp, fraudsters operate in highly coordinated bursts, creating dense clusters of spam within the network graph. When the proposed method performs its aggregation (averaging a nodes features with its neighbors), a spam review is mathematically combined with the surrounding spam reviews in its cluster. Instead of washing out the anomaly, the aggregation acts as an amplifier, concentrating the malicious signals and making the fraud glaringly obvious.

On Amazon, however, this exact same aggregation mechanism becomes the model's critical flaw. Fraudulent users on Amazon do not form dense, isolated spam clusters; rather, they are solitary actors embedded within massive crowds of normal shoppers. Because a fraudsters sharp, anomalous behavior is surrounded entirely by benign neighbors, the model's neighborhood aggregation actively destroys the fraud signal. It forcefully averages the sharp anomaly with the overwhelming benign noise of the surrounding users, blending the fraudster into the normal baseline and causing the proposed method to fail completely.

Consequently, a distinct strength of the proposed method lies in its powerful neighborhood aggregation mechanism, which makes it highly specialized for detecting coordinated fraud campaigns and dense clusters of malicious activity. By actively averaging and combining the features of neighboring nodes, the model acts as a powerful signal amplifier for clustered attacks, such as the synchronized spam rings frequently found on Yelp. Instead of treating each fraudulent action in isolation, the aggregation mathematically concentrates the shared behavioral traits of the entire local network. This means that even if an individual bot's behavior is subtle, the combined, aggregated signal of the dense spam cluster becomes glaringly obvious. While this architectural choice inherently means the model is less suited for identifying the solitary, dispersed anomalies present in datasets like Amazon, it establishes the proposed method as an exceptionally robust tool for uncovering organized, network-level fraud where malicious actors coordinate and connect.

All in all, the proposed RWR-GNN has implemented a lightweight random-walk-based neighbor strategy that inherently prioritizes informative neighbors and equilibrates long-range dependencies, offering a scalable and cohesive solution for fraud detection in heterogeneous and imbalanced graphs. Unlike GraphConsis Liu et al. (2020) or CARE-GNN (Dou et al., 2020), it eschews heuristic or reinforcement-intensive neighbor selection. Unlike PC-GNN Liu et al. (2021) and DOS-GNN (Jing et al., 2024), it does not depend exclusively on oversampling or dual aggregation; rather, it utilizes RWR to inherently weight informative neighbors according to walk probabilities. RWR-GNN is positioned as a scalable and theoretically robust solution that integrates neighbor refinement with fraud detection. Unlike PC-GNN Liu et al. (2021) and DOS-GNN Jing et al. (2024), which mitigate class imbalance via sampling or oversampling, the model has diminished imbalance effects by consistently reinforcing minority nodes during propagation through random-walk weighting, thereby circumventing the potential for synthetic noise or structural distortion. Furthermore, FRAUDRE (Zhang et al., 2021), RioGNN (Peng et al., 2021), and RWR-GNN offers a lightweight yet robust alternative that effectively captures both homophilic and heterophilic signals without considerable computational burden. By integrating these strengths, RWR-GNN provides a scalable, equitable, and resilient architecture that concurrently addresses heterophily, class imbalance, and noisy neighbor connections, attaining exceptional performance across various fraud detection contexts.

## 7   Conclusion

The reviewing systems witness fraudsters attitude to act as normal users to bypass anti-fraud systems, distribute fake information, or steal the private information of end users. The paper proposes the framework RWR-RGCN to enhance the fraud detection for the Yelp dataset and solve the challenges in baseline models by working with random walks with restart and community detection to expand the representation learning of the model to detect the fraud pattern; then the multi-RGCN classifier is used to predict the binary classification. The proposed framework enhances AUC and recall, as the main challenge is the processing power or hardware required to run the experiments, which could enhance the model results. The suggested model achieved 82.58% AUC, the highest recall (92%94%), and competitive AUC and G-Mean scores, giving a better prediction than the baseline. Various random walks show notable properties in the experiments. The study concludes that multi-relation graph fraud detection is important. RGCN results are heavily influenced by node selection random walks. In the future research path we will emphasize combining LLM-based representations with graph-based models to improve multimodal fraud detection.

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
