# OpenReview forum: "RWR-RGCN : A Novel Framework for Fraud Detection via Node Context Aggregation"
_TMLR — Rejected by TMLR_

### Review · Reviewer_NAQN · 2026-01-24

**Summary Of Contributions:**

This paper proposes RWR-RGCN, a fraud detection framework for multi-relational graphs that integrates Random Walk with Restart (RWR), Louvain community detection, and Relational Graph Convolutional Networks (RGCN). The motivation is to improve robustness against camouflage fraud, heterophily, and class imbalance by (i) using RWR to select informative neighbor sequences, (ii) aggregating node behavior at the community level via modularity-based clustering, and (iii) performing relation-aware message passing with RGCN. Experiments on the Yelp review fraud benchmark demonstrate improvements over several baselines, especially in recall, with reported AUC up to 82.58% and recall up to 94.56%.

Strengths:
1. The paper is well-motivated by known challenges in graph-based fraud detection, including camouflage behavior, heterophily, and class imbalance.
2. On the Yelp dataset, the proposed method achieves competitive or better AUC than strong baselines such as RioGNN and DOS-GNN, while substantially improving recall.

Weaknesses:
1. The paper does not provide ablations isolating the contribution of RWR, Louvain clustering, and their combination. It is therefore unclear how much each component contributes to the final performance gains.
2. The evaluation scope is narrow. All experiments are conducted on a single dataset (Yelp). Given the strong claims about robustness to heterophily and camouflage, additional datasets (e.g., Amazon, financial fraud graphs) would significantly strengthen the empirical evidence.
3. Results are reported as single numbers without considering statistical significance and variance.

**Audience:**

No

**Audience Explanation:**

The research question is important, but the findings are poorly supported and unconvincing.

**Claims And Evidence:**

No

**Claims Explanation:**

1.  The paper gives strong claims about robustness to heterophily and camouflage, but all experiments are conducted on a single dataset (Yelp), which is too narrow.

2. The proposed framework has multiple components, but the paper does not provide ablations isolating the contribution of RWR, Louvain clustering, and their combination. It is therefore unclear how much each component contributes to the final performance gains.

**Requested Changes:**

1. Try additional datasets (e.g., Amazon, financial fraud graphs).
2. Add ablation studies isolating the contribution of each component of the proposed framework.

---

> ### Author Response · Authors · 2026-04-18
> **Official Rebuttal Response to Reviewer NAQN**
>
> We thank the reviewer for their thoughtful and constructive feedback. We have revised the manuscript accordingly and address each point in detail below.
>
> **Q1:** Try additional datasets (e.g., Amazon, financial fraud graphs).
>
> **A** We appreciate the reviewer's comment on our experimental validation's scope. While showing our method on a single dataset may seem limited due to computational constraints, we appreciate the chance to clarify this choice and address the recommended alternative datasets. However, we took the reviewer's concern seriously and did further Amazon dataset testing as indicated. We added the justification for our results on Amazon in the discussion section.
>
> **Q2:** Add ablation studies isolating the contribution of each component of the proposed framework.
>
> **A** We appreciate the reviewer for raising this concern, which we concur are vital for justifying the design. A new 5.5 Ablation Study section has been incorporated into the amended manuscript.

---

### Review · Reviewer_8TWy · 2026-02-05

**Summary Of Contributions:**

The paper aims to address the problem of fraudulent review detection in online reviews. The approach adopted by authors is a relatively simple approach which combines Random Walk, Community Detection to help Relational GCNs perform fraud detection for multi-relational data. In doing so the method avoids relatively more complex approaches taken by peer works in the area involving Reinforcement Learning and Sampling. Key ideas behind their approach are a depth wise Random Walk based feature selector to help catch camouflage patterns, A community detection based aggregator to summarize node behavior and a Relational GCN feature learner built downstream from it. The authors try to establish these claims by comparisons across some baselines, but the choice of baselines is limited along with the datasets.

**Audience:**

No

**Audience Explanation:**

The issues explained above which revolve around result robustness, improper evaluation in terms of datasets and baselines reduce the confidence of a reader about the effetiveness of the approach and whether the justifications given for performance make sense beyond the single very specific dataset demonstration performed in the paper. Results established by the authors don't provide meaningful insights via ablations or analysis or relevant comparisons. Thus I am currently answering no to this question.

**Claims And Evidence:**

No

**Claims Explanation:**

The claims made by the authors w.r.t. accuracy and clarity of evidence are severely lacking. In terms of reported results in Table 3, the authors do not focus on establishing any helpful statistical grounding of the work. Even though authors show a marked improvement in Recall of their method over  the given baselines, the statistical difference between AUC characteristics between a few of those baselines, for eg. RioGNN, is very unclear and requires more investigation.
The method is demonstrated on just 1 dataset, making its utility unconvincing at best. The authors need additional datasets like Amazon, YelpChi, DGraph or CIS, to validate the claims about benefits of RWR and community detection based feature generation for fraud detection.
Another imperative that must be established is a comparison with more relevant baselines. The authors compare their method with GCN and graphSAGE but forego any comparison with more specialized baselines like [1-3] for instance. Though the scope of the work centers around GCN based frameworks, it would be helpful if the author could also comment on why they avoid any LLM based fraud review detection baselines?


### References
[1] DGA-GNN: Dynamic Grouping Aggregation GNN for Fraud Detection\
[2] Beta Wavelet Graph Neural Network for Fraud Detection.\
[3] Label-Exploring Graph Neural Network for Accurate Fraud Detection\

**Requested Changes:**

In addition to the inadequate results, which I hope the authors can improve upon markedly, the authors need to significantly rewrite large sections of the paper including figures (they could be significantly compressed) that demonstrate key results. An instance of poor writing can be found in Section 5 where Paragraph 2 and 3 are essentially clones of one another. Other writing issues are around verbose and unclear explanations in Sec 3 main body, and subsection 3.2.2. where the first paragraph explaining louvain's algorithm and the intuition in using it must be better described. I can clarify any questions the authors have w.r.t to rewriting after they add additional results as explained earlier.

---

> ### Author Response · Authors · 2026-04-18
> **Official Rebuttal Response to Reviewer 8TWy**
>
> We thank the reviewer for their thoughtful and constructive feedback. We have revised the manuscript accordingly and address each point in detail below.
>
> > **Q1:** The claims made by the authors w.r.t. accuracy and clarity of evidence are severely lacking. In terms of reported results in Table 3, the authors do not focus on establishing any helpful statistical grounding of the work. Even though authors show a marked improvement in Recall of their method over the given baselines, the statistical difference between AUC characteristics between a few of those baselines, for eg. RioGNN, is very unclear and requires more investigation.
>
> **A** We acknowledge the observation from the reviewer and we amended the AUC and the recall with RioGNN
>
> > **Q2:** The method is demonstrated on just 1 dataset, making its utility unconvincing at best. The authors need additional datasets like Amazon, YelpChi, DGraph or CIS, to validate the claims about benefits of RWR and community detection based feature generation for fraud detection.
>
> **A** We appreciate the reviewer's comment on our experimental validation's scope. While showing our method on a single dataset may seem limited due to computational constraints, we appreciate the chance to clarify this choice and address the recommended alternative datasets. However, we took the reviewer's concern seriously and did further Amazon dataset testing as indicated. We added the justification for our results on Amazon in the discussion section.
>
>
> > **Q3:** Another imperative that must be established is a comparison with more relevant baselines. The authors compare their method with GCN and graphSAGE but forego any comparison with more specialized baselines like [1-3] for instance. Though the scope of the work centers around GCN based frameworks
>
> **A**  We thank the reviewer for directing us to these three important works, which we had not included in our original baseline comparison. We have carefully read DGA-GNN, BWGNN, and LEX-GNN. In the revised manuscript we agree to modify section 5.4 and a complete comparison of AUC and recall versus all three baseline techniques have been added in the Table 4.
>
> We also draw attention to the Recall metric, which is crucial in fraud detection, where avoiding false negatives is a top priority. AUC alone can be misleading under severe class imbalance, but Recall—the rate at which actual fraudsters are accurately identified—is more operationally important due to the direct financial and reputational implications of missed fraud instances. In practical deployment settings, a model with slightly lower AUC but much greater recall may be better.
>
> > **Q4:** it would be helpful if the author could also comment on why they avoid any LLM based fraud review detection baselines?
>
> **A** We thank the reviewer for the question, We focus on graph-based fraud detection algorithms since they dominate transaction network relational and structural interdependence modeling. LLM-based fraud review detection baselines are not needed in our case because it does not entail review or semantic text data. We agree that graph learning with LLM-based cues like textual fraud explanations or user-generated information is promising. We will emphasize this as a future research path added to conclusion section, combining LLM-based representations with graph-based models to improve multimodal fraud detection.
>
> > **Q5:** In addition to the inadequate results, which I hope the authors can improve upon markedly, the authors need to significantly rewrite large sections of the paper including figures (they could be significantly compressed) that demonstrate key results.
>
> **A** We thank the reviewer for flagging this point, we removed figure 3 which was the 8 figures for AUC against RWR and figure 5 which was training and validation loss for RWR
>
> > **Q6:** An instance of poor writing can be found in Section 5 where Paragraph 2 and 3 are essentially clones of one another.
>
> **A** We thank the reviewer for highlighting these points. In revised manuscript. After merging and reorganizing them into a single paragraph with the primary argument, we're confident it will improve design choice and reader understanding before the technical specifics arrive.
>
> > **Q7:** Other writing issues are around verbose and unclear explanations in Sec 3 main body, and subsection 3.2.2. where the first paragraph explaining louvain's algorithm and the intuition in using it must be better described. I can clarify any questions the authors have w.r.t to rewriting after they add additional results as explained earlier.
>
> **A** We acknowledge the reviewer for highlighting the convoluted and ambiguous paragraphs in section 3. In the updated manuscript, we have rewrite the main body of section 3 and subsection 3.2.2, which it become section 4 and subsection 4.1.2 respectively after all the reviewers handling comments.
>
> If there are any remaining concerns, We would be happy to provide additional details.

---

### Review · Reviewer_GbFP · 2026-04-04

**Summary Of Contributions:**

This paper proposes RWR-RGCN, a unified framework for fraud detection in multi-relational graphs. The core idea is to enhance node representations by combining three components:
(1) Random Walk with Restart for capturing informative node sequences and contextual signals,
(2) Louvain clustering for identifying community-level structures
(3) a Relation-aware Graph Convolutional Network for learning relation-specific representations and performing node classification.

By integrating local walk-based features, global community structures, and relational message passing, the framework aims to improve robustness against challenges such as camouflage behavior, heterophily, and noisy edges in fraud detection scenarios.

Pros:
- The paper identifies important issues such as camouflage behavior and heterophily, which are indeed central in graph-based fraud detection
- The integration of RWR, Louvain clustering, and RGCN is conceptually coherent and aligns with known limitations of standard GNN aggregation
- The use of community detectionprovides some level of interpretability, which is often lacking in GNN-based fraud detection systems

Cons:
- The evaluation is conducted only on Yelp dataset, which is insufficient to demonstrate generalization. For a framework claiming broad applicability to fraud detection, experiments on multiple datasets (like, Amazon, financial fraud, or other graph benchmarks) are expected.
- The problem definition appears later in the paper (Section 3), after substantial method discussion. Structurally, it would be more appropriate to introduce the formal problem setup earlier (like in Section 2) to improve logical flow and readability
- The introduction and related work sections do not clearly articulate how this method differs fundamentally from existing approaches (like, CARE-GNN, PC-GNN, RioGNN).
The novelty appears to be largely a combination of existing techniques rather than a clearly isolated new mechanism.
- The introduction contains extensive general background but lacks a concise problem-to-method transition. This makes it harder to quickly grasp the key contribution and technical novelty.

**Audience:**

Yes

**Audience Explanation:**

Yes, the topic of fraud detection on multi-relational graphs is of interest to a subset of the TMLR community, particularly those working on graph neural networks and anomaly detection.

**Broader Impact Concerns:**

No concern

**Claims And Evidence:**

Yes

**Claims Explanation:**

While the paper provides empirical results showing improvements over baselines on the Yelp dataset, the evidence is only partially convincing.

The evaluation is limited to a single dataset, which makes it difficult to assess the generalizability of the proposed framework. Given that the method claims broad applicability to fraud detection in multi-relational graphs, validation on multiple datasets is necessary to support such claims.

**Requested Changes:**

- The current evaluation is limited to the Yelp dataset, which is insufficient to support claims of general applicability. The authors should include experiments on additional datasets (like, Amazon review datasets, financial fraud datasets, or other public graph benchmarks) to demonstrate robustness and generalization.
- Ablation studies to isolate component contributions. The proposed framework integrates multiple components (RWR, Louvain clustering, and RGCN), but it is unclear which component contributes most to the performance gains. The authors should provide systematic ablation studies removing or modifying each component to justify the design choices.
Clearer positioning and novelty clarification
- The paper should more explicitly distinguish the proposed method from prior work (like, CARE-GNN, PC-GNN, RioGNN). In particular, the authors should clarify whether the contribution is primarily methodological novelty or an effective integration of existing techniques, and explain why this combination leads to new capabilities.

---

> ### Author Response · Authors · 2026-04-18
> **Official Rebuttal Response to Reviewer GbFP**
>
> We thank the reviewer for their thoughtful and constructive feedback. We have revised the manuscript accordingly and address each point in detail below.
>
> >  **Q1:** The problem definition appears later in the paper (Section 3), after substantial method discussion. Structurally, it would be more appropriate to introduce the formal problem setup earlier (like in Section 2) to improve logical flow and readability
>
> **A**  We thank the reviewer for raising this concern. In the revised version, we established a dedicated Section 2 titled “Preliminaries and Problem Formulation” that introduces the formal graph definition, node and edge types, and the fraud detection task before any method discussion. The previous Section 3 titled “Background and Related Work” has been repositioned accordingly. This change required no new content — only reorganization of existing material.
>
> > **Q2:** The introduction and related work sections do not clearly articulate how this method differs fundamentally from existing approaches (like, CARE-GNN, PC-GNN, RioGNN). The novelty appears to be largely a combination of existing techniques rather than a clearly isolated new mechanism.
>
> **A** We thank the reviewer for the insightful suggestion. In the revised manuscript we agree to make some targeted changes to the introduction and related work that we believe it will enhance the readability of the manuscript. We have substituted the unclear final two paragraphs of 2.3 with an explicit differentiation paragraph that compares RWR-RGCN against CARE-GNN, PC-GNN, RioGNN, and DOS-GNN across five categories — camouflage handling, heterophily, class imbalance, community structure, and RL-free training. We have also included Table 1, a capability comparison matrix, which clearly highlights the design deficiencies. The core novelty of RWR-RGCN is its unique capability addressing all five fraud detection challenges simultaneously without requiring reinforcement learning or synthetic oversampling.
>
> > **Q3:** The introduction contains extensive general background but lacks a concise problem-to-method transition. This makes it harder to quickly grasp the key contribution and technical novelty.
>
> **A**  We express our gratitude to the reviewer for the observations provided. In the amended article, we agree to some modifications to the introduction, which we believe it will improve the manuscript's readability.
> For the problem-to-method transition concern: we have removed the first three paragraphs of the introduction, and substituted them with a two-sentence targeted motivator. We have added a dedicated transition paragraph at the end of the introduction.
>
> > **Q4:** The current evaluation is limited to the Yelp dataset, which is insufficient to support claims of general applicability. The authors should include experiments on additional datasets (like, Amazon review datasets, financial fraud datasets, or other public graph benchmarks) to demonstrate robustness and generalization.
>
> **A** We appreciate the reviewer's comment on our experimental validation's scope. While showing our method on a single dataset may seem limited due to computational constraints, we appreciate the chance to clarify this choice and address the recommended alternative datasets. However, we took the reviewer's concern seriously and did further Amazon dataset testing as indicated. We added the justification for our results on Amazon in the discussion section.
>
> > **Q5:**  Ablation studies to isolate component contributions. The proposed framework integrates multiple components (RWR, Louvain clustering, and RGCN), but it is unclear which component contributes most to the performance gains. The authors should provide systematic ablation studies removing or modifying each component to justify the design choices. Clearer positioning and novelty clarification.
>
> **A**  We appreciate the reviewer for raising this concern, which we concur are vital for justifying the design. A new 5.5 Ablation Study section has been incorporated into the amended manuscript.
>
> > **Q6:**  The paper should more explicitly distinguish the proposed method from prior work (like, CARE-GNN, PC-GNN, RioGNN). In particular, the authors should clarify whether the contribution is primarily methodological novelty or an effective integration of existing techniques, and explain why this combination leads to new capabilities.
>
> **A** We thank the reviewer for this concrete and significant suggestion. We address it directly. RWR-RGCN's is an integration contribution. RWR, Louvain, and RGCN are proven methods. We believe this integration is non-trivial and creates new capabilities for the following reasons. We added two paragraphs to the amended manuscript in the Introduction and two in the Related Work in subsection 3.2's closing paragraphs. and Table 1 state that the contribution prompted by a structural gap that creates a new capacity that no previous technique has achieved alone or together.

---

### Decision · Action_Editor_abDG · 2026-05-22

**Recommendation:** Reject

**Audience:**

Yes

**Audience Explanation:**

The paper will attract some audiences in graph learning and fraud detection.

**Claims And Evidence:**

No

**Claims Explanation:**

The paper addresses important challenges in graph-based fraud detection and presents a conceptually coherent integration of RWR, community detection, and RGCN. However, all reviewers still have substantial concerns regarding insufficient empirical validation, the absence of ablation analysis, and evaluation restricted to a single dataset with incomplete baseline comparisons. Overall, reviews are generally negative, and this work does not convincingly support the broad claims in the manuscript.